# Antibody-Drug Conjugates in Urothelial Carcinoma: A New Therapeutic Opportunity Moves from Bench to Bedside

**DOI:** 10.3390/cells11050803

**Published:** 2022-02-25

**Authors:** Antonio Ungaro, Marcello Tucci, Alessandro Audisio, Lavinia Di Prima, Chiara Pisano, Fabio Turco, Marco Donatello Delcuratolo, Massimo Di Maio, Giorgio Vittorio Scagliotti, Consuelo Buttigliero

**Affiliations:** 1Department of Oncology, University of Turin, San Luigi Gonzaga Hospital, Orbassano, 10124 Turin, Italy; antonio.unga@gmail.com (A.U.); alessandro.audisio93@gmail.com (A.A.); lavinia.diprima@unito.it (L.D.P.); chiara.pisano1991@gmail.com (C.P.); turcofabio9@gmail.com (F.T.); donatello.m.delcuratolo@gmail.com (M.D.D.); giorgio.scagliotti@unito.it (G.V.S.); 2Department of Medical Oncology, Cardinal Massaia Hospital, 14100 Asti, Italy; marcello.tucci@gmail.com; 3Department of Oncology, University of Turin, A.O. Ordine Mauriziano, 10124 Turin, Italy; massimo.dimaio@unito.it

**Keywords:** urothelial carcinoma, antibody-drug conjugates, ADC, Enfortumab vedotin, ADC resistance mechanism

## Abstract

Significant progress has been achieved over the last decades in understanding the biology and mechanisms of tumor progression in urothelial carcinoma (UC). Although the therapeutic landscape has dramatically changed in recent years with the introduction of immune checkpoint inhibitors, advanced UC is still associated with rapidly progressing disease and poor survival. The increasing knowledge of the pathogenesis and molecular pathways underlying cancer development and progression is leading the introduction of target therapies, such as the recently approved FGFR inhibitor Erdafitinib, or the anti-nectin 4 antibody drug-conjugate Enfortumab vedotin. Antibody drug conjugates represent an innovative therapeutic approach that allows the combination of a tar get-specific monoclonal antibody covalently conjugated via a linker to a cytotoxic agent (payload). UC is a perfect candidate for this therapeutic approach since it is particularly enriched in antigen expression on its surface and each specific antigen can represent a potential therapeutic target. In this review we summarize the mechanism of action of ADCs, their applications in localized and metastatic UC, the main mechanisms of resistance, and future perspectives for their use in clinical practice.

## 1. Introduction

Urothelial carcinoma (UC) is the sixth most frequent cancer in adults, with an estimated 573,258 new cases and 212,536 deaths worldwide in 2020 [1]. Risk factors include male sex, tobacco use, elderly age, and professional exposure to carcinogenic solvents. UC five-year survival is higher than 90% in non-muscle invasive bladder carcinoma (NMIBC). This percentage drops to 50 to 82% in muscle invasive bladder carcinoma (MIBC), while the five-year survival rate of metastatic patients is approximately 5%. About 75% of patients present NMIBC at diagnosis, a disease localized to the mucosa (Ta or carcinoma in situ) or invading the lamina propria (T1) [2]. Non-muscle invasive disease can be treated by transurethral resection, eventually followed by adjuvant local therapies, such as intravesical Bacillus Calmette-Guérin (BCG) or chemotherapy, depending on the relapse risk [3,4]. Unfortunately, in about half of high-risk patients, intravesical BCG treatment fails and NMIBC persists or recurs early. These patients do not benefit from further BCG instillations and current guidelines recommend early radical cystectomy (RC) as the backbone of treatment [4]. Recently, immunotherapy with pembrolizumab, an anti-Programmed cell Death protein 1 (PD1) monoclonal antibody, showed interesting results in the phase II trial KEYNOTE-057, conducted in patients with BCG-unresponsive NMIBC who were not eligible or declined RC, obtaining FDA approval [5].

MIBC occurs in 25% of patients with UC, and requires a multimodality approach due to the high rate of recurrence and metastatic progression [2]. The standard of care is currently represented by RC performed, in eligible patients, after cisplatin-based neoadjuvant chemotherapy [6]. Metastatic UC is a rapidly progressive tumor with a high mortality rate and limited therapeutic options. The management of metastatic UC has dramatically changed in the nineties with the introduction of the platinum-based chemotherapy, firstly with the use of the combination of methotrexate, vinblastine, doxorubicin, and cisplatin (MVAC), even in a dose-dense regimen and then, after the conduction of a head-to-head randomized trial, with the more tolerable association of cisplatin and gemcitabine [7]. For patients who are cisplatin-ineligible, treatment options include both chemotherapy and immunotherapy. EORTC 30986 trial demonstrated that carboplatin and gemcitabine is a reasonable treatment option in cisplatin-unfit metastatic UC patients [8,9]. Pembrolizumab and Atezolizumab are approved in PD-L1 positive cisplatin-ineligible patients or in patients who are not eligible to any platinum-containing chemotherapy [10]. Recently, Avelumab received approval as maintenance therapy in patients whose disease responded to first-line platinum-based chemotherapy with a median OS benefit of seven months compared to best supportive care [11]. Patients who progress to first-line treatment can receive immune checkpoint inhibitors (ICIs) if not previously administered in the first-line setting and regardless of PD-L1 expression, with response rates ranging from 15 to 20%, albeit with some long-term responses [12]. Subsequent line therapies may include taxanes and vinflunine, with limited efficacy in terms of response rates and disease control [13,14]. Despite the recent advances in the treatment of UC, both traditional chemotherapy and ICI have failed to generate long-term responses or ORR greater than around 30% in metastatic UC. Therefore, other treatments with higher efficacy are needed. In recent years, the increasing knowledge of pathogenesis and molecular pathways underlying UC progression are leading to the development of target therapies. In 2020, a consensus committee developed six subclasses of urothelial cancer with different molecular characteristics, prognosis, and response to therapy, representing an innovative approach to personalized medicine [15]. In particular, The Cancer Genome Atlas (TCGA) identified targetable mutations such as fibroblast growth factor receptor (FGFR), Human Epidermal Growth Factor Receptor 2 (HER-2), and Phosphatidylinositol-4,5-Bisphosphate 3-Kinase Catalytic Subunit Alpha (PIK3CA) [16]. Based on this evidence, Erdafitinib, an FGFR inhibitor, was tested in advanced urothelial cancer showing significant anti-tumor activity [17]. The efficacy of this drug was demonstrated in a phase II trial, including 99 patients with urothelial carcinoma eligible for second-line treatment after a platinum-based chemotherapy. The study reached the primary endpoint of overall response rate (ORR), achieving 40% of response rate [17]. Based on these significant results, the Food and Drug Administration (FDA) granted Erdafitinib accelerated approval as second-line in UC. In addition, multiple evidences show that UC is particularly enriched in antigen expression on its surface and each specific antigen can represent a potential therapeutic target14. Consequently, many studies have been conducted with a new category of therapeutic agents, known as antibody drug conjugates (ADCs), able to deliver chemotherapy drugs to a specific target with greater therapeutic efficacy and less toxicity [18]. In this review we describe the mechanism of action of main ADCs tested in UC, the biological and clinical rationale of their use in UC, the results obtained in both metastatic and localized disease, and the future perspectives about the potential development of these drugs alone and in combination with other therapies in order to overcome resistance mechanisms and personalize treatments.

## 2. Antibody-Drug Conjugates

ADCs are monoclonal antibody (mAb) covalently linked to small molecule anticancer agents. The antibody targets specific antigens selectively expressed on the tumor surface [19], delivering the chemotoxic payload locally and selectively to tumor cells. Some chemotherapeutic drugs cannot be used as classical chemotherapeutic agents, due to their poor handling or excessive toxicity [20]. The most relevant advantage of ADCs is represented by the ability to deliver selectively different types of cytotoxic agents to the tumor target increasing efficacy and reducing toxicity. Due to the unique mechanism of action, sophisticated technologies of developments are required to realize ADCs. The first approaches in the development of this class of drugs included the use of murine antibodies conjugated with cytotoxic agents such as doxorubicin, vinblastine, and methotrexate, despite their strong immunogenicity, poor selectivity, and limited potency [21]. With the evolution of more sophisticated technologies and the use of humanized antibodies, these agents have become more specific and effective and characterized by lower immunogenicity and higher potency [22]. ADCs consist of three components: an antibody binding specifically the antigen, a connecting linker, and a cytotoxic payload. These components vary greatly between different ADCs as a result of the intrinsic properties of the molecule and the interactions between them [23] (Figure 1).

### 2.1. Identification of a Suitable Antibody

Unlike other unconjugated mAbs, antibodies forming an ADC complex are not required to have an effector function and to elicit an immune response following link with the payload [24]. Over time, efforts have been made to investigate new antibody bases to improve therapeutic opportunities, such as fragments and bispecific antibodies. Currently, the cornerstone of structure is characterized by an immunoglobulin G [25]. Immunoglobulin G consists of four subclasses (IgG1, IgG2, IgG3, and IgG4) and differs from each other prevalently in the constant domain and hinge regions [26,27]. IgG1 is the most commonly used subtype in most engineered immunotherapies including ADCs due to its ability to stimulate immune effector function. Other advantages of IgG1 are its high serum stability (approximately 21 days half-life), low molecular weight, and wide distribution within the intra- and extravascular compartment [22]. IgG1 and IgG4 are more stable than IgG2 due to hinge region and the presence of disulfide bonding. IgG3 have a short elimination half-life, causing a higher risk of immunogenicity [28,29]. Ideal mAb selection should consider the identification of an antigenic target strongly expressed on malignant cells and not expressed in those non-malignant. The specific selection of target is crucial to deliver the cytotoxic payload throughout cells expressing it, enhancing therapeutics window and reducing systemic toxicities. Further aspects to be considered are a limited antigen immunogenicity and cross-reactivity, as well as a strong binding affinity towards the target to allow effective internalization and stability [18,30]. In addition, the drug-to-antibody ratio (DAR) is crucial for ADC activity. A very low DAR negatively affects its potency, while a very high DAR has a negative impact on pharmacokinetics [31]. Indeed, the conjugation of a mAb with doxorubicin or monomethyl auristatin E (MMAE), forming an ADC with high DAR, produces a compound with higher hydrophobicity, higher levels of aggregation, and increased clearance. High DARs allow ADC greater internalization with increased efficacy but, conversely, may lead to increased clearance [32,33,34]. Using ADCs with limited immunogenicity allows an important advantage in therapeutic values, since the activation of the immune response may lead the development of anti-drug antibodies that can decrease or suppress the efficacy of drug itself [35,36,37]. The first attempts of ADC with murine mAbs showed serious limitations such as low selectivity and poor penetration into tumor cells at the expense of a significant immunogenicity and systemic toxicity. Based on the above, the use of IgG subclasses is intended not only to bind to the target, but also to influence the distribution and efficacy of an ADC. In this regard, an effective and personalized selection is mandatory to obtain a potentially effective ADC [38].

### 2.2. Linker

The function of the linker is to bind the antibody to the cytotoxic payload through the conjugation site located in the antibody heavy chains. Linkers have to satisfy two fundamental characteristics. The first one is to ensure that the cytotoxic payload remains safely bound to the antibody, particularly during circulation in the plasma. Indeed, if the linker is unstable, the cytotoxic payload could be released prematurely into the plasma circulation, resulting in systemic toxicity and lack of therapeutic efficacy [39]. This aspect is crucial when ADCs deliver drugs that cannot otherwise be delivered systemically [40]. The second characteristic is the ability of the linker to deliver the drug within the tumor context [41]. Linkers are divided into two major subclasses: cleavable and non-cleavable. Cleavable linkers have the property of breaking down and releasing the cytotoxic payload of ADCs due to factors present in the tumor microenvironment [42]. There are three different mechanisms that can lead to a cleavable breakage process. The first mechanism is driven by glutathione, strongly represented in the cytoplasm compared to the extracellular compartment, allowing the breakage of disulfide bonds and the release of the cytotoxic payload. A further feature of these linkers is to confer better solubility to the ADC than dipeptide linkers. This mechanism is used in ADCs, such as indatuximab ravtansine and mirvetuximab soravtansine [38,40]. The second mechanism is provided by linkers that cleave in acidic pH environments such as Hydrazone. These linkers have the ability to exploit the acidic environment of endosomes and lysosomes for hydrolysis. Premature cleavage of the linker and release of the payload into the circulation results in the onset of hepatotoxicity described in gemtuzumab ozogamicin [39,43,44]. 

Protease-dependent linkers have the peculiarity of being degraded by lysosomal proteases by recognition of specific peptide sequences and subsequent hydrolysis. The particular characteristic of these linkers allows a greater stability of the ADC within the plasma (which has different pH), avoiding a premature release of the payload. Some examples of using these linkers are Sacituzumab govitecan and Enfortumab Vedotin [45].

Non-cleavable linkers are more stable than cleavable linkers. Their mechanism of action relies on the degradation of the entire antibody-linker construct to release the payload. An example of non-separable linkers are T-DM1 and belantamab mafodotin [46].

### 2.3. Payloads

Cytotoxic agents that constitute the payload of ADCs are generally heavily toxic molecules [47]. Specifically, as these molecules cannot be delivered alone, they require to be carried by specific antibodies to a specific target. The first ADCs tested were capable of delivering traditional chemotherapeutic agents, such as doxorubicin, methotrexate, and vinca alkaloids [48,49,50]. However, these agents did not demonstrate greater efficacy compared to standard chemotherapeutic agents, requiring high doses of drug, with non-negligible risks of systemic toxicities [51]. Some evidence has also shown that only a very small amount of antibody directed against the tumor reaches the tumor tissue, suggesting the utilization of drugs with very high cytotoxic power at nanomolar or lower concentrations [47,52,53]. Furthermore, limitation to the cellular permeability of the payload, the engineering of linkers, and the selection of effective targets of the mAb allow the non-occurrence of off-target drug in terms of premature complex cleavage. However, it is useful to consider that the reduced solubility of the complex severely impairs a neighbor-response phenomenon termed the "Bystander Effect". This process allows neighboring cells which lack the target (because not expressed or inhibited by other mechanisms) to experience the therapeutic effects indirectly, via diffusion and cell signaling mechanisms [54,55].

The most important payload macro categories are agents that destabilize microtubules, drugs capable of generating DNA damage and protein toxins.

Microtubule destabilizers, which include auristatins and maytansins, are derived from natural bacteria. The auristatins include monomethyl auristatin E (MMAE) and monomethyl auristatin F (MMAF), which are synthetic derivatives of the dolastatin 10 peptide derived from Dolabella Auricularia [56]. These drugs inhibit tubulin polymerization leading to cell cycle arrest and apoptosis. Similarly, maytansins such as DM1, originated from benzoansamacrolides and derivatives, target tubulin via the vinca alkaloid binding site resulting in blockade of mitotic replication, cell cycle arrest, and apoptosis. Emtansine trastuzumab is an example of this category of drugs, formed by DM1 and trastuzumab through a non-cleavable linker [57,58].

Payloads that act directly on DNA damage, such as calicheamicin, cuocarmicins, and pyrrolobenzodiazepines, have the function of generating DNA double helix damage and its peculiarity is to be not specific cell cycle agents, rather they act as alkylating agents. They are able to disrupt the transcription sequence, causing DNA double helix breakage and subsequent apoptosis [59,60,61,62]. Other drugs included in this macro-category are camptothecin analogues, such as the exatecan derivative (Dxd) and the active metabolite of irinotecan, SN-38. Both are able to inhibit topoisomerase I with consequent DNA damage and breakage [63,64].

Finally, protein toxins are another class of payload and are used as immunotoxins. This category is structurally similar to ADCs despite comprising an antibody or its fragment connected to a protein toxin through a fusion gene. This category is intended to inhibit protein synthesis through a mechanism of damage on ribosomal RNA. Several clinical studies are in progress on this category including as payload the diphtheria toxin, the exotoxin A of Pseudomonas Aeruginosa, or saporins [65,66,67,68,69,70].

## 3. ADC Anti-Tumor Activity in Urothelial Carcinoma

### 3.1. Metastatic Setting

#### 3.1.1. Enfortumab Vedotin

Enfortumab vedotin is an ADC developed to target nectin-4, conjugated to a MMAE Enfortumab vedotin is an ADC developed to target nectin-4, conjugated to a MMAE payload via a protease-cleavable linker [71]. Nectin-4 is a type I transmembrane polypeptide member of the nectin family encoded by the NECTIN4 gene, that is widely expressed and associated with poor prognosis in metastatic UC. In physiological conditions, human nectin in physiological conditions is specifically enriched in the placental and embryonic tissues but its expression significantly decreases in adult life [72,73]. Generally, normal tissues, such as the skin, bladder, salivary glands, esophagus, and stomach, have low or mostly moderate expression of nectin-4 [74]. The main function of nectin-4 is to allow proper adhesion of cells junctions held together with cadherins. Elevated expression of nectin-4 has been observed in several tumors, including bladder, breast, lung, and ovarian cancer [75]. Multiples evidences showed that the promotion of metastatic process through the WNT beta-catenin and PI3K-AKT-mTOR signaling pathways, as well as the interaction with the ERBB2 tyrosine kinase receptor is closely related to nectin 4 expression [76]. EV was firstly studied in patients with metastatic urothelial cancer previously treated with chemotherapy as part of the phase I study EV101 [77]. The encouraging results of this study suggested the design of a phase II study testing the dose of 1.25 mg per kg on days 1, 8, and 15 every 28 days. The EV-101 study enrolled 155 patients, including 112 who received the recommended dose for phase II. All patients received at least one platinum-containing chemotherapy and 79% at least one ICI. The most frequent side effects (AEs) were fatigue (53%), alopecia (46%), loss of appetite (42%), dysgeusia (38%), nausea (38%), sensory peripheral neuropathy (38%), pruritus (35%), diarrhea (33%), and maculopapular rash (21%). Grade 3 and 4 toxicities, including hyperglycemia (5%), were uncommon. Four deaths related to ongoing treatment were described, including respiratory failure, urinary tract obstruction, diabetic ketoacidosis, and multi-organ failure. The ORR was 43% with a median progression free survival (PFS) of 5.4 months and a median overall survival (OS) of 12.3 months in patients treated with the dose of 1.25 mg per kg [77]. The open-label, single-arm phase II trial (EV-201) evaluated the efficacy of EV in patients pretreated with immunotherapy enrolled in two cohorts: cohort 1 enrolled patients previously treated with platinum-containing therapy; and cohort 2 platinum-ineligible patients [78]. The ORR in cohort 1 (125 patients) was 44% and the median duration of response (mDOR) was 7.6 months. The estimated median PFS was 5.8 months and the median OS was 11.7. Responses were observed in all subgroups, including patients unresponsive to ICI. Adverse events (AEs) seen in more than 20% of patients included fatigue (50%), alopecia (49%), rash (48%), loss of appetite (44%), peripheral sensory neuropathy (40%), and dysgeusia (40%). The EV-301 study, a randomized, open-arm phase III trial of EV versus investigator-choice chemotherapy (docetaxel, paclitaxel, and vinflunine) enrolled 608 patients progressing after platinum-containing chemotherapy and ICI. The study reached its primary endpoint, obtaining a median OS of 12.9 and 9.0 months respectively for EV and chemotherapy (hazard ratio (HR): 0.70; 95% confidence interval (CI) 0.56 to 0.89; *p* = 0.001) [79]. Based on these data, EV received FDA and EMA approval for the treatment of patients previously treated with platinum-containing chemotherapy and ICI [80]. In first-line setting, EV has also been shown to have a synergistic effect when combined with ICI in the cohort A of the ongoing phase Ib/2 EV-103 trial, showing an ORR of 73%, with 15.6% of CR and median PFS of 12.3 months, in cisplatin-unfit patients [81] (Table 1).

In January 2022, a warning has been issued on treatment with EV concerning severe adverse skin reactions and pneumonia. The drug has also been warned for hyperglycemia, pneumonitis, peripheral neuropathy, ocular disorders, infusion-site extravasation, and embryofetal toxicity [82].

In January 2022, a warning has been issued on treatment with EV concerning severe adverse skin reactions and pneumonia. The drug has also been warned for hyperglycemia, pneumonitis, peripheral neuropathy, ocular disorders, infusion-site extravasation, and embryofetal toxicity [79].

#### 3.1.2. Sacituzumab Govitecan

Sacituzumab Govitecan (SG) is an ADC conceived to specifically target human trophoblastic cell surface antigen 2 (Trop-2). SG consists of a monoclonal antibody against Trop-2 conjugated to SN-38, an active metabolite of irinotecan, through a hydrolysable linker with a DAR of 7.6 [63,80].

Trop-2 is a transmembrane glycoprotein first identified on trophoblastic cells. High expression of Trop-2 has been found in several epithelial tumor tissues, correlating with higher tumor aggressiveness and worse prognosis [81]. In UC, Trop-2 overexpression is closely related with increased disease aggressiveness [82]. 

SG was initially tested in a phase I dose escalation trial including 25 patients with different tumor types showing an acceptable profile of toxicity and an encouraging therapeutic activity [83]. In a phase I/II study 45 patients with metastatic urothelial carcinoma who progressed after ≥1 prior systemic therapy were treated with SG at 10 mg per kg on days 1 and 8 of 21-day cycles, until progression or unacceptable toxicity [78]. The ORR was 31% with a clinical benefit rate of 47%. The median DOR was 12.6 months. The mPFS and mOS were 7.3 and 18.9 months, respectively. The most common grade 3 or higher reported side effects were neutropenia (38%), anemia (11%), hypophosphatemia (11%), diarrhea (9%) fatigue (9%), and febrile neutropenia (7%). The TROPHY-U-01 study is an open-label, single-arm phase II study evaluating the efficacy of SG in patients progressing after platinum-containing chemotherapy and a checkpoint inhibitor. Preliminary data from cohort 1, including 113 patients with locally advanced or unresectable or metastatic UC who had progressed after prior platinum therapy and ICI, showed an ORR of 27% with a mPFS and mOS of 5.4 months and 10.9 months, respectively. The observed grade greater than 3 AEs were neutropenia (35%), leukopenia (18%), anemia (14%), diarrhea (10%), and febrile neutropenia (10%). These preliminary data support accelerated approval of SG in this setting [84].

#### 3.1.3. Sirtratumab Vedotin (ASG15-ME)

Sirtratumab vedotin (SV) is an ADC able to specifically target SLITRK6, a member of the neuronal transmembrane protein family [87,88]. Overexpression of this protein has been observed in several tumor types including bladder, lung, breast, and glioblastoma. Lack of SLITRK6 has been reported in patients with sensorineural deafness and myopia [89].

SV is composed of a human gamma 2 antibody selectively directed against SLITRK-6 and conjugated to a payload composed of MMAE held together through a protease-cleavable linker with a DAR of 4 [88]. SLITRK6 expression is not homogeneous on tumor cells with 5% of cells negative for this protein. Despite this evidence, it was observed that both SLITRK-6 negative tumor cells and cells expressing low levels of SLITRK6 respond to SV treatment [88]. First data on SV anti-tumor activity were reported in a phase I study enrolling 51 patients with metastatic urothelial cancer were reported the first data on SV anti-tumor activity. SLITRK6 expression was determined by immunohistochemistry in about 93% of patients. In 42 patients treated with a therapeutic dose (>0.5 mg per kg) an ORR of 33% was observed. The mDOR was 15 weeks with an mPFS of 16 weeks. The most common grade 3 or higher AEs was fatigue (44%). Reversible ocular toxicities were observed in 10 patients, none of whom experienced grade 3 toxicity [90]. Currently, there are no ongoing trials evaluating the SV efficacy in UC metastatic setting [87].

#### 3.1.4. Targeting HER-2 in Bladder Cancer

The tyrosine kinase receptor erbB-2, better known as HER-2, is a member of the epidermal growth factor receptor (EGFR) family of tyrosine kinase receptors. In case of HER-2 overexpression, heterodimerization of this receptor with other tyrosine kinase receptors belonging to the EGFR family leads to activation of signaling pathways promoting cells proliferation and tumorigenesis [90]. A wide spectrum of therapies targeting HER-2 have demonstrated significant activity in patients with HER-2 positive breast cancer [91,92]. In UC 12% of tumors have HER-2 overexpression [93]. HER-2 alterations are more frequent in luminal variants (clusters I and II) than in basal variants of UC (clusters III and IV) [94]. Some data suggest that Her-2 overexpression in MIBC is related with higher tumor aggressiveness and worse prognosis [95].

Considering these data, the activity of some ADCs against HER-2 cancer was evaluated in metastatic bladder.

##### 3.1.4.1. Trastuzumab Emtansine (TDM-1)

TDM-1 is an ADC which combines the anti-tumor activity of trastuzumab, an antibody directed against Her-2, with a payload formed by emtansine (DM-1), an anti-microtubule agent. The antibody and the cytotoxic agent are covalently linked via a stable linker, to specifically release the anti-microtubule drug in cells expressing HER2 [96]. 

The unique mechanism of action of this drug and the wide use in HER-2 positive breast cancer in various settings represent the rationale to hypothesize a role also in urothelial cancer.

Although preclinical models showed that TDM-1 is able to deeply inhibit the growth of bladder cancer cell lines, a phase II basket trial did not show a significant activity of this drug in patients with metastatic urothelial carcinoma [97,98]. The results of the phase II KAMELEON trial (NCT02999672), which evaluates the efficacy of TDM-1 in UC, cholangiocarcinoma and pancreatic cancer, are pending.

##### 3.1.4.2. Trastuzumab Deruxtecan (DS-8201a)

Trastuzumab Deruxtecan (DS-8201a) is an ADC consisting of a monoclonal antibody targeting HER-2 conjugated to a topoisomerase I inhibitor (DXd) at a DAR of 7–8. This ADC has shown significant activity even in tumor cells expressing low levels of HER-2 [67]. In early trials conducted in heavily pretreated metastatic breast cancer patients DS-8201 showed high response rate [99]. A phase II trial evaluating the efficacy of DS-8201 in several tumors including metastatic UC is currently ongoing (NCT04482309). Some preclinical data have also shown the role of DS-8201 in the immunogenic modulation of the tumor microenvironment. Trials testing the association of DS-8201 and ICIs such as Nivolumab are ongoing to evaluate the safety and efficacy of combinations [100].

##### 3.1.4.3. Disitamab Vedotin (RC-48)

Disitamab vedotin (previously known as RC-48) is a novel ADC consisting of a humanized monoclonal antibody directed against HER-2 conjugated to MMAE through a cleavable linker with a DAR of 4 [101]. A recent phase II study reported encouraging results in 43 patients with HER-2^+^ metastatic urothelial cancer previously treated with at least one line of systemic treatment platinum-based chemotherapy. The ORR was 51%, mPFS and mOS were 6.9 and 13.9 months, respectively [102]. Another phase II trial enrolling a larger population is starting to test the efficacy of this agent in HER-2^+^ UC metastatic patients (NCT04879329) (Figure 2).

##### 3.1.4.4. Drug-Conjugates beyond Antibodies

Although several attempts have been made in recent decades to effectively target HER-2 using different approaches, such as antibodies and tyrosine kinase inhibitors, the occurrence of resistance mechanisms and the presence of side effects require the development of additional targeting approaches. Peptide-drug conjugates (PDCs) are a novel and investigational class of pro-drugs that selectively deliver a payload via a sequence-specific peptide that binds to, or within, the tumor surface. Due to the small size of the peptide, PDCs also reduce the possibility of undesirable immunogenic effects and are biodegradable. In addition, their low molecular weight allows them to be highly purified using high-performance liquid chromatography (HPLC). These molecules can be subdivided on the basis of their characteristics into cell-targeting and cell-penetrating PDCs. Cell-targeting PDCs are designed to selectively bind to receptors present on the surface or on tumor vascular endothelial cells [103]. Among them, hybrid peptides seem to play an important role in targeting HER-2. In particular, Karasseva et al. have identified a hexapeptide (KCCYSL) very frequently involved in a population of phages that is affinity-selective against the extracellular domain of HER-2 [104]. This evidence led to the possibility of using dedicated software (e.g., Molecular Dynamics) to elaborate similar peptide sequences useful for generating molecules with higher affinity for HER-2. As suggested by the work of Birì-Kovacs et al., the synthesis of new higher-affinity peptides can be obtained by starting from peptides already known to have an affinity for HER-2, resulting in different peptide analogues [105]. The use of these peptides has both diagnostic and therapeutic implications, and allows the use of innovative payloads, such as the use of metal-organic complexes as anti-tumour [106].

### 3.2. Activity of ADC in Localized Bladder Cancer

#### Oportuzumab Monatox

Oportuzumab monatox (OM) is a recombinant fusion protein consisting of a humanized single variable chain fragment binding an epithelial cell adhesion molecule (EpCAM) fused to Pseudomonas exotoxin (ETA-252-608) [107]. EpCAM is a transmembrane protein which plays a key role in cell adhesion and survival signaling. EpCAM represents an interesting target due to its overexpression in epithelial tumors such as lung, colon, ovary, breast, prostate, and pancreatic cancer [108]. Moreover, EpCAM is deeply expressed in cancer stem cells resistant to anticancer agents [109]. However, EpCAM is also expressed in many physiological epithelial tissues, severely limiting its use as an ADC due to its toxicity [110]. The mechanism of action of this drug is almost overlapping with other ADCs: it is internalized after binding to EpCAM and releases the exotoxin, inducing apoptosis [111]. This agent is administered locally to limit systemic toxicities and is being studied in patients with NMIBC refractory to intravesical BCG. A phase I trial enrolled 64 patients refractory or intolerant to intravesical BCG therapy with high-grade urothelial cancer and stage Tis, Ta or T1. The drug was administered weekly for 6 weeks followed by 12-week reassessment. The maximum tolerated dose was not reached. All patients were able to safely complete the six scheduled cycles, without treatment-limiting toxicities. Approximately 39% of patients [24] had a CR [112]. These encouraging data were confirmed by the subsequent phase II trial enrolling 46 patients whose disease was refractory or intolerant to BCG. The trial included two treatment arms treated with 30-mg weekly instillation of 6 (cohort 1) and 12 weeks (cohort 2), respectively, followed by three tri-weekly maintenance cycles every 3 months. CR was achieved in 44% of patients, 16 % of patients maintained a CR at 1-year follow-up. Although CR rates were similar in both arms, the median time to disease recurrence was longer in cohort 2 [113]. The open-label, non-randomized phase IIl study (VISTA study) enrolled 133 BCG refractory NMIBC patients. The primary endpoint was the rate of CR at 1 year, which was 39% in cohort 1 enrolling patients with time to disease recurrence < 6 months after BCG, 80% in cohort 2 enrolling patients with recurrence between 6 and 11 months, and 68% in cohort 3 enrolling patients with papillary tumors that recurred within 6 months of BCG [114].

A combination trial with the anti PL-L1 Durvalumab is currently ongoing in patients with high-grade NMIBC refractory to intravesical BCG (NCT03258593).

## 4. Potential Mechanism of Resistance

Nowadays, resistance mechanisms associated with the use of ADCs in UC are poorly understood. A more accurate understanding of these mechanisms may provide additional insights regarding the drug intrinsic mechanisms of action and may accelerate the development of predictive biomarkers of efficacy. Early evidences suggests that ADCs resistance can occur due to several mechanisms including the prevention of antibody attachment, alteration of ADC processing and internalization, and loss of payload efficacy [41].

### 4.1. Lack of Antigen Attachment

A frequent barrier to the long efficacy of ADC is loss of target antigen within the tumor, that can occur by downregulation of the antigen gene expression, by gene mutations rendering the antigen less recognizable, or by selection of non-antigen-expressing cells within a highly heterogeneous cells population [115]. The loss or reduction of expression of the antigen carried by the ADC may result in both loss of antibody binding and release of payload. Coates et al. reported an association between response to SG treatment and loss of Trop-2 expression in patients with metastatic triple negative breast cancer [116]. In the EMILIA trial which led to the approval of TDM-1 in HER-2-positive metastatic breast cancer the benefit of TDM-1 was greater in patients expressing high levels of HER-2 mRNA than in patients expressing low levels of HER-2 mRNA [117]. Similarly, in the ASCENT trial, patients treated with SG in metastatic triple-negative breast cancer have benefited more from treatment in case of Trop-2 expression high or moderate, compared with low or absent expression [66].

Further studies are needed to clarify whether this mechanism of resistance may be overcome by multimodal approaches, such as the use of bispecific antibodies to reach multiples targets.

### 4.2. Suppression of Payload Efficacy

A common mechanism of chemotherapy resistance is the elimination of the drug from the cell microenvironment via ATP-binding cassette (ABC) transporters [118]. These efflux transporters confer resistance to ADCs because most cytotoxic agents used as payloads are substrates of ABC transporters [119,120]. Some preclinical data showed that auristatin analogs and maytansinoids are substrates of drug transporters such as multidrug resistance-1 (MDR-1). Prolonged exposure to these agents selects cell clones that overexpress the MDR-1 transporter [121].

Another mechanism of resistance may arise due to the occurrence of resistance mutations on the molecular target of payload. The occurrence of resistance mutations in Topoisomerase-1 (TOP1) which is the molecular target of SN-38 may correlate with decreased susceptibility to SG treatment [116].

### 4.3. Cell Cycle Alterations

The cell cycle plays a key role both in tumorigenesis and in establishing novel resistance mechanisms. Some evidences showed that TDM-1-resistant cells undergo increased expression of cyclin B [122].

In addition, alterations in apoptosis regulation may interfere with the efficacy of ADCs. The most commonly altered pathways involve protein regulation of BAX and BAK and overexpression and mutation of BCL-2 and BCL-X, as observed in patients exposed to Gemtuzumab ozogamicin [123].

### 4.4. Alteration in Trafficking Pathways

In order to optimize the efficacy of ADCs, internalization of the antibody into the cell by endocytosis is required. Notably, this process can occur through clathrin-mediated, caveolin-mediated, and clathrin–caveolin-independent endocytosis processes [124]. However, these mechanisms may also limit the efficacy of payload itself. Some preclinical studies have shown that internalization of TDM-1 into caveolin-1 coated vesicles correlates with decreased sensitivity to treatment and increased insensitivity to treatment [125].

## 5. Role of ADCs in the Therapeutic Sequence of Advanced UC

ADCs represent nowadays a new therapeutic opportunity for patients with advanced UC. Due to the recent approval of ICIs and FGFR-inhibitors in the same disease setting clinicians face with several challenges. The most important issue is the choice of correct treatment sequence in each patient in order to personalize treatment and maximize anti-neoplastic efficacy. Cytotoxic-based chemotherapy remains the backbone of first-line therapy for cisplatin-eligible and for cisplatin-ineligible patients. In addition, immunotherapy can be a therapeutic option for PD-L1 positive platinum eligible and all platinum-ineligible patients. Immunotherapy is also indicated as maintenance after platinum-based chemotherapy and in second line treatment. In patients progressing on immunotherapy treatment, EV or FGFR inhibitors can be taken into account, according to FGFR status. Patients progressing to platinum, not already treated with ICIs, can be treated with Erdafitinib if FGFR positive or, if negative, with ICIs. In the choice of treatment, we need to consider clinical characteristics of neoplastic disease, patients’ comorbidities, time to progression, prior lines of treatment, and the FGFR status. In patients with a higher burden of disease or symptomatic, EV or FGFR inhibitors are the preferable options compared with the immunotherapy, due to their higher percentages of response and faster time of response. Conversely, recent evidence showed that immunotherapy is associated with a higher rate of durable responses. 

## 6. Future Perspectives

Despite the rapidly expanding therapeutic horizons of ADCs, new clinical and translational strategies are needed in order to maximize treatment potential of ADCs. The intrinsic mechanism of action of this drug class has the great advantage of delivering cytotoxic agents with powerful antitumor activity. The immune modulation of the tumor microenvironment has the great advantage of operating in a system broader than the single cell. Therefore, cytotoxic agents possess a synergistic activity when combined with ICIs. Cytotoxic agents are able to promote cell death, causing release of tumor antigens. These effects allow the activation of the immune system and the increase of antigen-presenting cells. Moreover, ICIs counteract the immunosuppression generated in the tumor microenvironment by modulating regulatory T cells, immunosuppressive cytokines, and enzymes with immunomodulatory function [126].

The combination of chemotherapy and ICIs is a new weapon already approved in several cancers and new combinations between different drug classes are in clinical trials. 

With the advent of new agents, such as ADCs, and novel combination approaches the scenario of therapeutic options in bladder cancer is rapidly expanding [127].

Preclinical studies have investigated the ability of ADCs to modulate the immune system. Gardai et al. showed that ADCs holding MMAE as payload induce immune cell death (ICD) and can stimulate anti-tumor immunity [128]. ICD is characterized by the induction of the endoplasmic reticulum stress response and surface presentation of damage-associated molecular patterns (DAMPs) of the immunomodulatory system. These DAMPs, mostly represented by Toll-like receptors, can perform their immune action previously suppressed by the tumor microenvironment. In addition, subsequent studies in mouse models have shown that tumor regression was greater with the combination of Brentuximab vedotin and a PD1 inhibitor, confirming the therapeutic synergy of these two drugs. Based on these data, it is reasonable to assume that the combination of ADC and ICIs in vivo may result in a strong anti-tumor response [129].

Several trials are underway to investigate the safety and efficacy of the combination of ICI and ADC. The EV-103 trial is a multi-arm phase Ib/II trial evaluating the efficacy of EV alone or in combination with pembrolizumab and/or chemotherapy in patients with locally advanced or metastatic bladder cancer. Preliminary results have shown that EV in combination with Pembrolizumab in the first-line setting (cohort A) achieves an ORR of 73.3% with an mPFS of 12.3 months and an unreached mDOR [130]. The focus of the combination of ADC and ICIs has also shifted to early settings. A phase III randomized open-label trial in cisplatin-fit patients to receive EV+ Pembrolizumab in the perioperative setting vs. neoadjuvant therapy with cisplatin + gemcitabine is currently ongoing [131]. The TROPHY U-01 study is an ongoing phase II multi-cohort trial testing the efficacy of SG in metastatic bladder cancer after failure of a first-line therapy. Results from these studies are not currently available [85].

With regard to the development of novel ADC agents, efforts are directed towards understanding and overcoming primary and acquired resistance mechanisms and reducing toxicities. Several agents targeting different targets, such as EGFR, integrin β6, B7-H1, and CD25, are currently under investigation in early phase basket trials (Table 2). Table 2 describes the ongoing trials investigating ADC. 

## 7. Conclusions

ADCs represent an innovative concept of administration of conventional cytotoxic therapies, allowing a reduction in toxicities and an enhancement of therapeutic activity. ADCs could be particularly effective in neoplastic diseases characterized by specific and targetable antigens. Their use is particularly appealing in neoplastic diseases with limited therapeutic possibilities. These drugs represent currently a viable option in the therapeutic landscape of metastatic or locally advanced UC progressing to chemotherapy. Besides EV already approved in this setting, several other ADCs are being investigated, including SG. Future researches will have to investigate the role of ADCs in early stages of disease and the efficacy of combinations with other innovative agents in order to maximize efficacy and overcome mechanisms of resistance.

## Figures and Tables

**Figure 1 cells-11-00803-f001:**
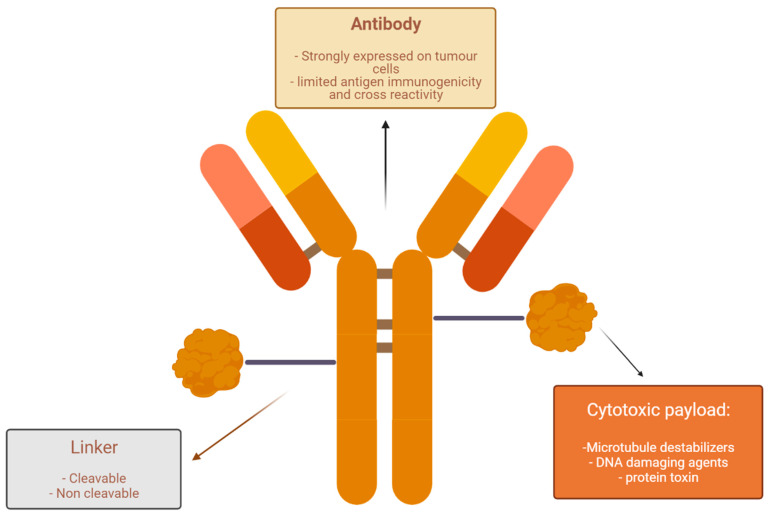
Structure and main functions of antibody-drug conjugate components.

**Figure 2 cells-11-00803-f002:**
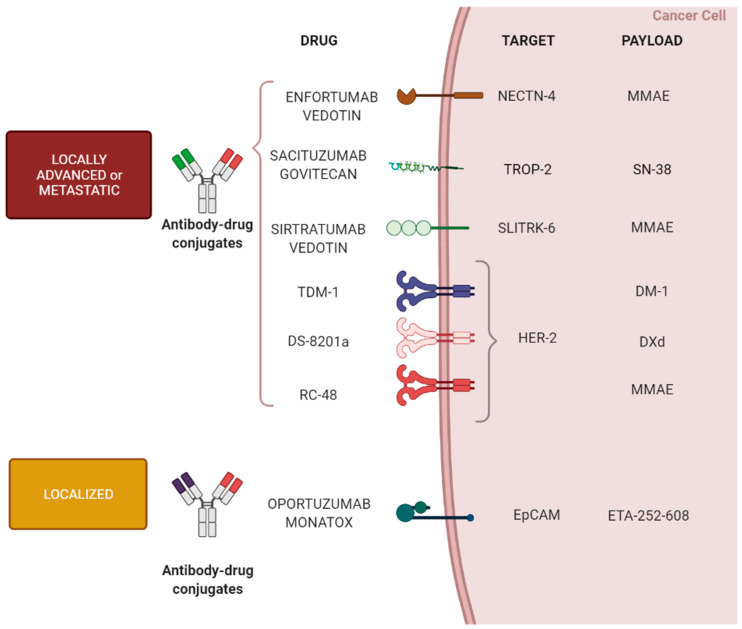
The main mechanisms of antibody drug conjugates investigated in urothelial carcinoma.

**Table 1 cells-11-00803-t001:** Clinical Data on the use of ADCs in Urothelial Carcinoma.

Drug Name	Target	Cytotoxic Payload	Study	Trial Phase	Sample Size	ORR (%)	Median Progression Free Survival (mPFS) in Months (95% CI)	Median Overall Survival (mOS) in Months (95% CI)	Median Duration of Response (mDoR) in Months	Adverse Events (G3–G4)
**Enfortumab Vedotin**	Nectin-4	MMAE	Rosemberg et al. [74]	1	155	43	5.4 (5.1–6.3)	12.3 (9.3–15.3)	7.4 (5.6–9.6)	34%
			Rosemberg et al. [Cohort 1] [75] Y. Yu et al. [Cohort 2] [82]	2	125 89	44 52	5.8 (4.9–7.5) 5.8 (5.03–8.28)	11.7 (9.1-not reached) 14.7 (10.51–18.20)	7.6 (0.95–11.30) 6 (2.8–8.3)	54% 55%
			Powles et al. [76]	3	608	40.6 vs. 17.9	5.5 vs. 3.7 (HR, 0.62; 0.51–0.75; *p* < 0.001)	12.8 vs. 8.9 (HR, 0.70; 0.56–0.89; *p* = 0.001)	5.0 (0.5–19.4)	51% vs. 48%
**Sacituzumab Govitecan**	Trop-2	SN-38	Bardia et al. [83]	1/2	45	28.9	6.8 (3.6–9.7)	16.8 (9.0–21.9)	12.9 (3.8–22.5)	59%
			Tagawa et al. [84]	2	113	27	5.4 (3.5–7.2)	10.9 (9.0–13.8)	7.2 (4.7–8.6)	not evaluable
**Sirtratumab Vedotin**	SLITRK-6	MMAE	Petrylak et al. (interim analysis) [85]	1	51	33	4	-	3.9	50%
**Disitamab Vedotin**	HER-2	MMAE	Sheng [86]	2	43	51.2%	6.9 (5.6–8.9)	13.9 (9.1–NE)	6.9 (4.7–10.8)	58%

Monomethyl auristatin E (MMAE); not enriched (NE); and human epidermal growth factor receptor 2 (HER2).

**Table 2 cells-11-00803-t002:** Current Ongoing Trials for ADCs in Urothelial Cancer.

NCT Number and Study Name	Drug Name	Setting	Phase	Study Characteristics	Recruitment Status
NCT03288545 EV-103	Enfortumab vedotin	Metastatic	I/II	Safety and anticancer activity of Enfortumab vedotin (EV) given intravenously as monotherapy and in combination with other anticancer therapies as first line (1L) and second line (2L) treatment for patients with urothelial cancer. The primary goal of the study is to determine the safety, tolerability, and efficacy of Enfortumab vedotin alone and in combination with pembrolizumab and/or chemotherapy.	Recruiting
NCT04223856 EV-302	Enfortumab vedotin + Pembrolizumab vs chemotherapy	Metastatic	III	An Open-label, Randomized, study of Enfortumab Vedotin in Combination with Pembrolizumab Versus Chemotherapy Alone in Previously Untreated Locally Advanced or Metastatic Urothelial Cancer	Recruiting
NCT04225117 EV-202	Enfortumab vedotin	Metastatic	II	An Open-label, Multicenter, Multicohort, to Evaluate Enfortumab Vedotin in Subjects with Previously Treated Locally Advanced or Metastatic Malignant Solid Tumors	Recruiting
NCT04960709 VOLGA	Enfortumab vedotin + Durvalumab +/− tremelimumab	Perioperative	III	Randomized, Open-Label, Multicenter Study to Determine the Efficacy and Safety of Durvalumab in Combination with Tremelimumab and Enfortumab Vedotin or Durvalumab in Combination With Enfortumab Vedotin for Perioperative Treatment in Patients Ineligible for Cisplatin Undergoing Radical Cystectomy for Muscle Invasive Bladder Cancer	Recruiting
NCT03924895 KEYNOTE-905/EV-303	Enfortumab vedotin + Pembro vs. Pembro vs. surgery alone	Perioperative	III	A Randomized Study Evaluating Cystectomy with Perioperative Pembrolizumab and Cystectomy with Perioperative Enfortumab Vedotinand Pembrolizumab Versus Cystectomy Alone in Cisplatin-Ineligible Participants with Muscle-Invasive Bladder Cancer	Recruiting
NCT04700124 KEYNOTE-B15/EV-304	Enfortumab vedotin + Pembrolizumab vs. Cisplatin + Gemcitabine	Perioperative	III	A Randomized, Open-label Study to Evaluate Perioperative Enfortumab Vedotin Plus Pembrolizumab (MK-3475) Versus Neoadjuvant Gemcitabine and Cisplatin in Cisplatin-eligible Participants with Muscle-invasive Bladder Cancer	Recruiting
NCT04527991 TROPiCS-04	Sacituzumab govitecan vs. chemotherapy	Metastatic	III	A Randomized Open-Label Study of Sacituzumab govitecan Versus Treatment of Physician’s Choice in Subjects with Metastatic or Locally Advanced Unresectable Urothelial Cancer	Recruiting
NCT03547973 TROPHY-U-01	Sacitizumab govitecan	Metastatic	II	Open Label, Study of Sacituzumab govitecan in Metastatic Urothelial Cancer After Failure of Platinum-Based Regimen or Anti-PD-1/ PD-L1 Based Immunotherapy	Recruiting
NCT04482309 DESTINY-PanTumor02	Trastuzumab deruxtecan	Metastatic	II	Multicenter, Open-label Study to Evaluate the Efficacy and Safety of Trastuzumab Deruxtecan (T-DXd, DS-8201a) for the Treatment of Selected HER2 Expressing Tumors (DESTINY-PanTumor02)	Recruiting

## Data Availability

Not applicable.

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
