# Peer review of "Antibody-Drug Conjugates in Urothelial Carcinoma: A New Therapeutic Opportunity Moves from Bench to Bedside"

_cells, 2022, doi:10.3390/cells11050803_

Round 1

Reviewer 1 Report

The authors have written a well summarized review on ADC for Urothelial carcinoma. Although I would like to highlight some necessary modifications.

  1. As section Trastuzumab Emtansine (3.1.5), DS-8201a (3.1.6) and RC 48 (3.1.7) all fall under the section 3.1.4 (Targeting Her2 in bladder cancer), so I would suggest TDM-1 to be sub section under Her2 i.e. it should be 3.1.4.1, DS-8201a 3.1.4.2 and like wise.
  2. As Her2 targeting is so well described, the authors could also include some addtional information of targeting Her2 using peptide (may not be directly relevant but might be good for the discussion section to compare with antibody. For eg. the peptide KCCYSL and the following articles could be cited: Biomolecules. 2020 Feb; 10(2): 183, Breast Cancer Res 23, 84 (2021),  Materials Science and Engineering C 68 (2016) 327–337, Mol Pharm. 2011 Jun 6; 8(3): 901–912.,.

Author Response

Point 1

As you suggested, I have divided section 3.1.4 into additional sub-paragraphs.

Point 2

We thank the reviewer for this review. as you suggested, I have included an additional paragraph (3.1.4.4) explaining the potential therapeutic potential of peptide-drug conjugates in Her-2 positive disease, based on the bibliography you suggested. I have attempted to explain the main mechanism of action of these molecules. Their rationale, the strengths of these small molecules and their possible use also with innovative payloads have been described. 

Reviewer 2 Report

In this manuscript, "Antibody-drug conjugates in urothelial carcinoma: a new therapeutic opportunity moves from bench to bedside" Ungaro A. with coauthors discussed the approach for urothelial carcinoma treatment. The manuscript is clearly written and topic on a timely subject. The authors discussed in detail the mechanisms of action of antibody-drug conjugates, their applications, resistance, and perspectives in the future. I will suggest accepting the manuscript for publication in the present form. 

Author Response

We thank you for your comments and observations.

Reviewer 3 Report

This is a well-written review about antibody-drug conjugates in urothelial carcinoma. I will suggest it for publication after the following point are addressed.

  1. One recent review (doi.org/10.3390/ph13090245) related to antibody-drug conjugates should be included in this ms.
  2. The resolution of figure 2 should be improved to a higher level.
  3. In the section of 2.2. Linker, some typical cleavable linkers are encouraged to be discussed.
  4. Line 222, is 79 the number of a ref?

Author Response

Point 1

We thank the reviewer for the remark. As suggested, I have included the citation of the updated review you pointed out within the manuscript (cit. 44).

Point 2

We thank you for the observation. As you indicated, I have uploaded image 1 and 2 with a higher resolution.

Point 3

We thank you for your comment. As you suggested, a more thorough description of cleavable linkers has been included, describing the various subcategories to which they differentiate among themselves. In particular, for each subcategory we have tried to give typical examples of ADCs using that specific type of linker.

Point 4

We thank you for your comment. Due to a typing error, this numbering has appeared and does not refer to any citation. We have removed the incorrect numbering. We apologize for the inconvenience.
